# Morpho-Physiological Strategies of *Shorea leprosula* Miq. and *Shorea acuminata* Dyer in Response to Light Intensity and Nutrient Amendments

Abd Razak Siti Nurfaeiza [1], Wan Ahmad Wan Juliana [2], Khamis Shamsul [1,*] and Md. Nor Shukor [1]

[1] Department of Biological Sciences and Biotechnology, Faculty of Science and Technology, Universiti Kebangsaan Malaysia, Bangi 43600, Selangor, Malaysia

[2] Institute of Climate Change, Universiti Kebangsaan Malaysia, Bangi 43600, Selangor, Malaysia

* Correspondence: shamsulk@ukm.edu.my

**Abstract:** Successfully restoring degraded forest areas depends on seedlings adapting their growth to suit harsh environments. Hence, the requirements for seedlings' growth need to be addressed before replanting degraded sites. The present study determines the effect of abiotic factors viz. light irradiance (8%, 30%, and 100%), nutrient addition (no fertiliser (NF), NPK, and vermicompost) on the growth performance and photosynthetic capacity of two dipterocarp species seedlings, *Shorea leprosula* Miq. and *Shorea acuminata* Dyer. The morphological characteristics assessed for growth performance comprised plant height, stem diameter, leaf count, leaf area, relative chlorophyll concentration, biomass, and root-to-shoot ratio. Li-Cor 6400 and 6800 were used to measure the leaf gas exchange traits, including photosynthetic rate (A), transpiration rate (E), intercellular $CO_2$ concentration (Ci), stomatal conductance ($g_{sw}$), and water-use efficiency (WUE). Our results demonstrated that different levels of light intensity and nutrient amendment significantly impacted plant-growth performance. Plants grown in 30% irradiance showed better growth performance in terms of relative height growth rate (RHGR), mean number of leaves, and leaf areas 41%, 24%, and 32% higher than the control. The A value was also higher in 30% irradiance, but no significant differences were observed between each level of light irradiance. The addition of vermicompost gave better growth for RHGR, relative diameter growth rate (RDGR), mean number of leaves, biomass, and relative chlorophyll concentrations 47%, 40%, 131%, 19%, and 27% higher than the control, respectively. However, the results obtained for photosynthetic parameters were contrary to growth performance. The photosynthesis rate (A) was higher (14.8%) in NPK compared to the control, and the other photosynthetic parameters did not differ significantly despite different nutrient amendments. In terms of species, *S. leprosula* has better growth performance and photosynthetic characteristics than *S. acuminata* in different light irradiance and nutrient amendments, thereby rendering *S. leprosula* the preferred rehabilitation species. Generally, nutrient addition of either NPK or vermicompost and 30% light irradiance gave better morphological and physiological growth for both species. The outcome of this study could provide a better understanding on the forest rehabilitation strategy to reduce the seedling-mortality rate, particularly for climax tree species.

**Keywords:** dipterocarp; light irradiance; vermicompost; photosynthesis; forest rehabilitation; climate change

## 1. Introduction

Most of the lowland forests of Southeast Asia are dipterocarp forests [1–3] with Dipterocarpaceae the most dominant family over vast areas in Southeast Asia's forests [4]. It is an essential plant in Malaysia, not only for its valuable timber tree species but also for the presence of the genus Shorea [5]. *Shorea* Roxb. is the largest genus of Dipterocarpaceae and the most dominant emergent tree genus in tropical Asia [6]. There have been minimal conservation efforts of dipterocarps in recent decades as the family was regarded as a

common species free of threats. However, due to the changes in land-use patterns to meet the increasing demands of resources, the sustainability of dipterocarp species needs to be addressed.

Deforestation is defined as converting forests for other land uses [7]. Farming, agriculture, mining, urbanisation, and plantation are leading causes of deforestation. Between 2001 and 2018, Malaysia estimated that around 34.0% of total tree cover was lost due to deforestation [8]. Uncontrolled deforestation will lead to forest degradation, which lowers productivity and hampers the essential forest services. Forest rehabilitation was identified as a reliable effort to restore the forest area. The reduction of forest area as the primary terrestrial carbon sink disrupts the global carbon cycle and increases atmospheric carbon dioxide concentration. This reduction impacts global warming in two ways: the acceleration of greenhouse gas emissions, such as carbon dioxide, methane, and nitrogen oxides; and reduction in carbon sequestration by tropical trees, such as *Shorea* species, during photosynthesis [9]. Forest destruction has led to carbon emissions with 4.3–5.5 Gt $CO_2$ eq/year [10]. Thus, protecting and restoring this vast carbon sink is essential for mitigating climate change.

There is an urge to combat climate change and its impacts due to the increment of the 2020 global average temperature. The temperature was reported at 1.2 °C above the pre-industrial baseline which is woefully off track to stay at or below 1.5 °C, as stated in the Paris Agreement [11]. The Bonn Challenge and New York Declaration on Forest (NYDF) aim to restore 150 million hectares of degraded land by 2020 and 350 million hectares by 2030 [10]. In addition, these efforts were aligned with the existing international commitment, such as the Aichi target from the Convention on Biological Biodiversity [12], the UNFCC REDD+ goal [13], and the sustainable development goals (SDGs) by the United Nations [11]. In Malaysia, the rehabilitation efforts are actively executed through a tree-planting program by the government which is known as Greening Malaysia. To date, 46,298,820 trees from 1349 species were planted to achieved the goal of 100 million trees planted by 2025 [14].

Forest rehabilitation is arguably the most promising way to restore forest capacity. However, the selection of tree species should be prioritised to achieve the ultimate goals of rehabilitation. The use of native tree species rather than exotic tree species should be emphasised to preserve the well-known tropical rainforest dominant tree species. The use of native tree species could influence the structure and function of the tropical forest [5]. *Shorea* is a fast-growing hardwood and the most promising plant for dipterocarp plantation in Peninsular Malaysia [5,15–18]. The success of each rehabilitation effort differs based on site conditions. Factors that determine the success of rehabilitation include the type of disturbance, forest remnant, climate [19], seed dispersal [20], wood density [21], species selection, seedling quality [22], and site maintenance in the early stage of the establishment [7]. Nevertheless, the real determinant for restoration success is still a mystery as the success of each restoration differs based on the condition of the local environment [19].

Generally, various environmental conditions affect plant growth, such as light, soil water content, temperature, and soil nutrients [23]. Among these, light is the most important environmental factor affecting plant establishment, growth, and survival [24,25]. Taiz and Zeiger [26] also supported the importance of light in plant growth as light energy was used to produce ATP and NADPH in the light reactions of photosynthesis. The optimum light irradiance to support plant growth, especially in an altered environment, is scarce and varies depending on plant species. Additionally, the need for nutrient addition to the available nutrient content in soil needs to be addressed, especially in dipterocarp species, to achieve optimum growth. The present study aims to investigate the growth performance and photosynthetic responses of two *Shorea* species in different light irradiance and nutrient amendments.

## 2. Materials and Methods

### 2.1. Study Area

The study site is located at Tembat Forest Reserve (FR), consisting of Tembat and Puah Dams, north of the existing Kenyir Dam in Kuala Berang, Hulu Terengganu, Peninsular Malaysia (Figure 1). It is around 50 km from Gua Musang-Hulu Terengganu highway and around 65 km west of Kuala Terengganu. The estimated terrain elevation of Tembat FR is 284 m above sea level. Its latitude is 5°05′03″ N, and the longitude is 102°44′13″ E. Tembat FR is a part of the Central Forest Spine (CFS), running parallel to the Main Range. Its existence is crucial for biodiversity and environmental protection. The study site can be categorised as a lowland dipterocarp forest to a hill dipterocarp forest in which the altitudes range between 150 m and 420 m.

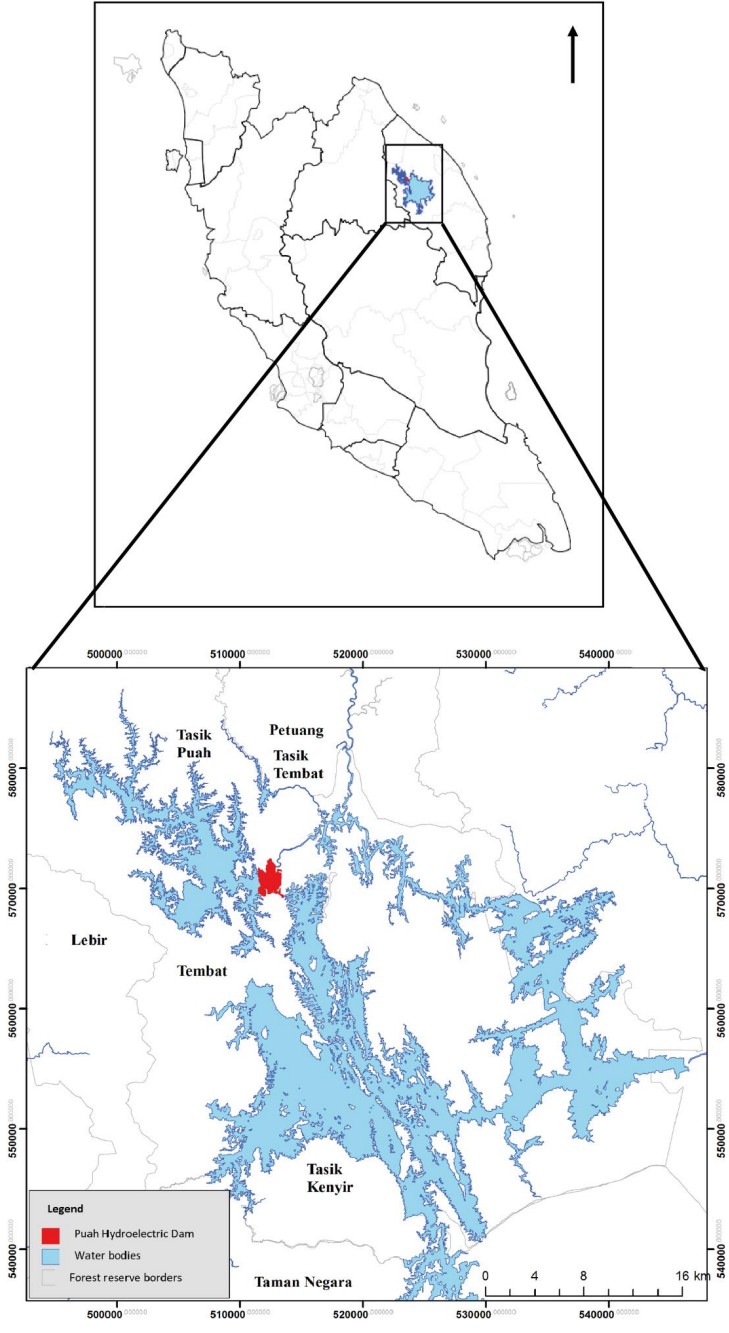

**Figure 1.** Study area, Tembat Forest Reserve, Terengganu, Peninsular Malaysia.

## 2.2. Plant Materials and Experimental Design

The experiment was conducted for seven months from October 2018 until May 2019 at Hulu Terengganu, Peninsular Malaysia, and the factorial RCBD design was implemented. In this study, both *Shorea leprosula* Miq. and *S. acuminata* Dyer were used because these species assemble good traits and have the survival potential that matches the rehabilitation sites. Nineteen-month-old seedlings of *S. leprosula* and *S. acuminata* were obtained from Ajil's Nursery owned by Forestry Department Terengganu. Three seedlings from both species were planted in a 914.4 cm$^2$ customised planter box (*n* = 3). Two treatments were applied: light irradiance and nutrient amendments with three replicates for each level of treatment. The levels of light irradiance were 100% (control), 30%, and 8%, while the nutrient applications were divided into the control (without fertiliser), NPK, and vermicompost (VC) (Figure 2).

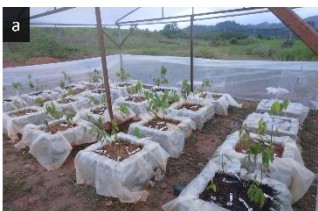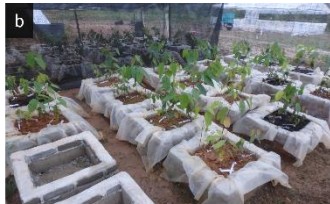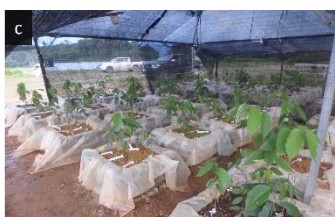

**Figure 2.** *Shorea leprosula* Miq. and *Shorea acuminata* Dyer tree seedlings in three levels of light irradiance; (**a**) 100% (control); (**b**) 30%; and (**c**) 8%.

## 2.3. Data Collection

### 2.3.1. Growth Parameters

The plant height, stem diameter, number of leaves, and relative chlorophyll concentration were measured monthly for seven months. Plant height was measured from the soil surface until the tips of the upper shoot using a ruler. Using a digital Vernier calliper, the stem diameter was measured 2 cm from the soil surface. A chlorophyll content meter (Hansatech Instruments Ltd., King's Lynn, Norfolk, UK) was used to measure relative chlorophyll concentration. At the end of the experiment, the shoots and roots of each plant were sampled and immediately dried at 80 °C in an oven for 72 h to measure dry weight and later the biomass accumulation. The leaf area was measured using Image-J software (Version 1.52a, Bethesda, MD, USA). Each young, matured leaf was photographed using a digital camera on a whiteboard with a customised 3 cm scale. Then, the images were processed using the software to obtain the leaf area.

### 2.3.2. Photosynthetic Parameters

Photosynthesis rate (A, µmol m$^{-2}$ s$^{-1}$), transpiration rate (E, mmol$^{-2}$ s$^{-1}$), stomatal conductance (g$_{sw}$, mol m$^{-2}$ s$^{-1}$), water-use efficiency (WUE, µmol CO$_2$ mmol$^{-1}$ H$_2$O), and intercellular carbon concentration (Ci, µmol mol$^{-1}$) were measured for the leaf gas exchange traits. All gaseous-exchange measurements were conducted with portable photosynthesis-measurement equipment (Li-6400XT and Li-6800; LI-COR, Inc., Lincoln, NE, USA) referring to Yong et al. [27] with some modification. The measurements were taken at an ambient CO$_2$ concentration of 400 µmol mol$^{-1}$, air temperature of 29–30 °C, relative humidity of 55–65%, and photosynthetic photon flux density (PPFD) of 1300 µmol mol$^{-1}$ in the leaf chamber. One fully expanded leaf per plant was selected for each of the measurements.

## 2.4. Data Analysis

The collected data were subjected to analysis of variance (ANOVA) using R Studio (Version 1.2.5033, Boston, MA, USA) to find out if there were any significant differences among the treatment levels of light intensity and nutrient application with a significant level of $p < 0.05$. When significant differences were found, the means were separated using Tukey's HSD test. Principal component analysis (PCA) was performed using morphological data and leaf gas exchange measurements to reduce data dimensionality.

## 3. Results

The growth performances for both species were the highest in 30% irradiance (Table 1) except for the biomass, relative chlorophyll concentration, and root-to-shoot ratio. The leaf gas exchange parameters showed the highest photosynthesis rate (A) and stomatal conductance ($g_{sw}$) in 30% irradiance (Table 2). In terms of the nutrient amendment, all growth traits showed higher responses when treated with vermicompost than the control except for leaf area and root-to-shoot ratio. The leaf area was higher with the amendment of NPK instead of vermicompost. The growth traits also gave significant differences among nutrient levels and species. Transpiration rate (E), water-use efficiency (WUE), and intercellular carbon concentration (Ci) have higher values with the application of vermicompost compared to the control, while photosynthesis rate (A) and stomatal conductance ($g_{sw}$) favoured the application of NPK (Table 2). Three out of five photosynthetic characteristics were also higher with vermicompost amendments compared to the control (no fertiliser).

### 3.1. Growth and Photosynthetic Responses of S. leprosula and S. acuminata Grown under Different Light Irradiance

Comparing three different levels of light irradiance, the control showed the highest photosynthetically active radiation (PAR) compared to 30% (single layer of shading) and 8% (double layer of shading) irradiance (Figure 3). The peak of light irradiance was at noon for all light-intensity levels. The control recorded the highest PAR which reached up to 1200 $\mu mol\,m^{-2}\,s^{-1}$ followed by 30% (400 $\mu mol\,m^{-2}\,s^{-1}$) and 8% (150 $\mu mol\,m^{-2}\,s^{-1}$) irradiance.

Generally, *S. leprosula* showed better growth performance in all growth parameters compared to *S. acuminata* except for the root-to-shoot ratio (Figure 4). *Shorea leprosula* showed the highest RHGR, RDGR, and the mean number of leaves in 30% light irradiance (Figure 4a–c). The RHGR and mean number of leaves for *S. leprosula* grown in 30% irradiance were doubled compared to the control. RDGR was 18.2% higher in 30% irradiance (0.30 $mm\,mm^{-1}\,month^{-1}$) than the control (0.25 $mm^{-1}\,month^{-1}$). In contrast, the highest biomass and relative chlorophyll concentration (RCC) were recorded in 8% irradiance with both 24.4% and 38.3% higher than the control, respectively. The root-to-shoot ratio was twofold lower in 8% irradiance than the control (Figure 4f).

**Table 1.** The effects of light irradiance and nutrient amendments on morphological growth characteristics in two *Shorea* species.

| Traits | Significance | | | | | | Means | | | | | |
| --- | --- | --- | --- | --- | --- | --- | --- | --- | --- | --- | --- | --- |
| | Species | Light Irradiance (LI) | Nutrient | Species × LI | Species × Nutrient | LI × Nutrient | 100% | 30% | 8% | NF | NPK | VC |
| RHGR (cm cm$^{-1}$ month$^{-1}$) | *** | *** | *** | ** | ns | ns | 0.056 | 0.079 | 0.064 | 0.051 | 0.073 | 0.075 |
| RDGR (mm mm$^{-1}$ month$^{-1}$) | *** | *** | *** | *** | ns | ns | 0.13 | 0.13 | 0.11 | 0.10 | 0.13 | 0.14 |
| Mean number of leaves (plant$^{-1}$ month$^{-1}$) | ** | ns | *** | *** | ns | ns | 2.94 | 3.65 | 2.90 | 2.02 | 2.96 | 4.67 |
| Leaf area (cm$^2$) | *** | *** | *** | *** | *** | ns | 43.8 | 57.6 | 55.4 | 43.3 | 58.5 | 54.9 |
| Biomass (g plant$^{-1}$) | *** | ns | ** | *** | ns | * | 6.42 | 6.28 | 6.96 | 5.98 | 6.59 | 7.12 |
| Root-to-shoot ratio | *** | *** | *** | * | ns | ** | 1.26 | 0.75 | 0.88 | 1.21 | 0.98 | 0.69 |

Note: RHGR = relative height growth rate; RDGR = relative diameter growth rate; NF = no fertiliser; VC = vermicompost; *, **, *** indicate significant at $p < 0.05$, $p < 0.01$, and $p < 0.001$, respectively; ns = non-significant.

*Shorea acuminata* showed different growth responses compared to *S. leprosula*. Four out of seven parameters for *S. acuminata* showed lower values in both 8% and 30% compared to the control viz. RDGR, means number of leaves, biomass, and root-to-shoot ratio (Figure 4). The RDGR and means number of leaves were two times and three times lower, respectively, in 8% irradiance than the control. Biomass did not show a significant difference in 30%

irradiance (4.95 g plant$^{-1}$) compared to the control (6.21 g plant$^{-1}$), while the root-to-shoot ratio was almost four times lower in 30% irradiance than the control. RCC doubled in 8% irradiance compared to the control (Figure 4g).

For all the leaf gas exchange characteristics, *S. leprosula* has greater values compared to *S. acuminata* (Figure 5), with a significant difference between these two species (Table 2). *Shorea leprosula* recorded the highest photosynthesis rate (A) in 30% light irradiance compared to the control (Figure 5a), but the differences were insignificant. The transpiration rate (E), stomatal conductance ($g_{sw}$), and water-use efficiency (WUE) showed the lowest values in 8% irradiance compared to the control (Figure 5b,d,e).

All the leaf gas exchange parameters for *S. acuminata* were higher in 30% irradiance than the control except for WUE and intercellular $CO_2$ concentration (Ci), which showed the opposite trend (Figure 5a–e). Surprisingly, *S. acuminata* has a significantly higher Ci value than *S. leprosula* despite different light irradiance levels.

**Table 2.** The effects of light irradiance and nutrient amendments on physiological characteristics in two *Shorea* species.

| Traits | Significance | | | | | | Means | | | | | |
|---|---|---|---|---|---|---|---|---|---|---|---|---|
| | Species | Light Irradiance (LI) | Nutrient | Species × LI | Species × Nutrient | LI × Nutrient | 100% | 30% | 8% | NF | NPK | VC |
| RCC (SPAD) | ** | *** | *** | ns | ns | ns | 4.65 | 6.04 | 8.00 | 5.27 | 6.65 | 6.69 |
| A ($\mu$mol m$^{-2}$ s$^{-1}$) | *** | ns | * | ns | ns | ns | 7.84 | 8.56 | 7.07 | 7.65 | 8.78 | 7.04 |
| E (mmol$^{-2}$ s$^{-1}$) | *** | ** | ns | ns | ns | ns | 0.0027 | 0.0028 | 0.0022 | 0.0027 | 0.0026 | 0.0024 |
| Ci ($\mu$mol mol$^{-1}$) | ns | ns | ns | ns | ns | ns | 282 | 217 | 216 | 247 | 221 | 249 |
| $g_{sw}$ (mol m$^{-2}$ s$^{-1}$) | *** | ** | ns | ns | ns | ns | 0.16 | 0.14 | 0.13 | 0.16 | 0.16 | 0.14 |
| WUE ($\mu$mol $CO_2$ mmol$^{-1}$ H$_2$O) | ns | ns | ns | ns | ns | ns | 0.00048 | 0.00049 | 0.00039 | 0.00042 | 0.00037 | 0.00058 |

Note: RCC = relative chlorophyll concentration; A = photosynthesis rate; E = transpiration rate; Ci= intercellular $CO_2$ concentration; $g_{sw}$ = stomatal conductance; WUE = water-use efficiency; NF = no fertiliser; VC = vermicompost; *, **, and *** indicate significant at $p < 0.05$, $p < 0.01$, and $p < 0.001$, respectively; ns = non-significant.

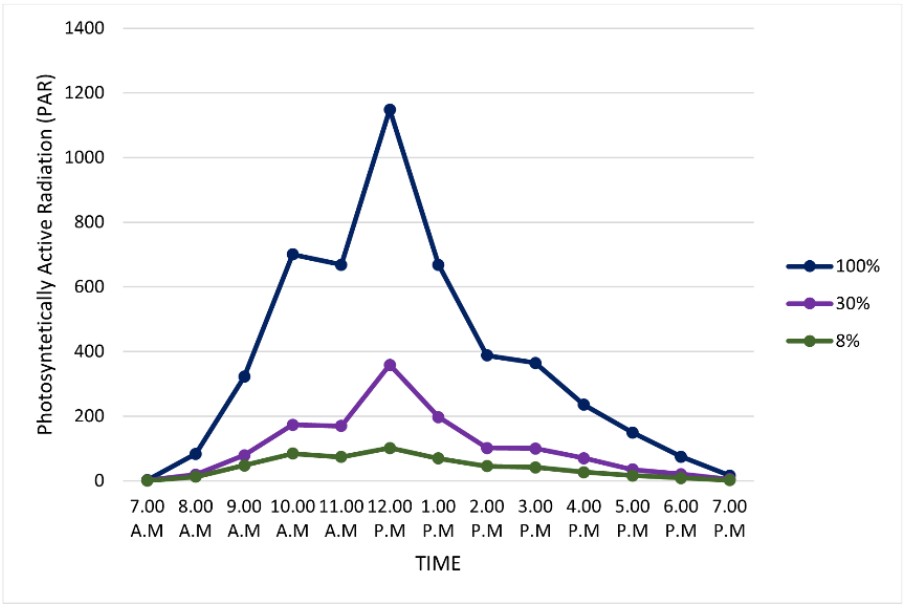

**Figure 3.** Photosynthetically active radiation (PAR) differences between three levels of light irradiance.

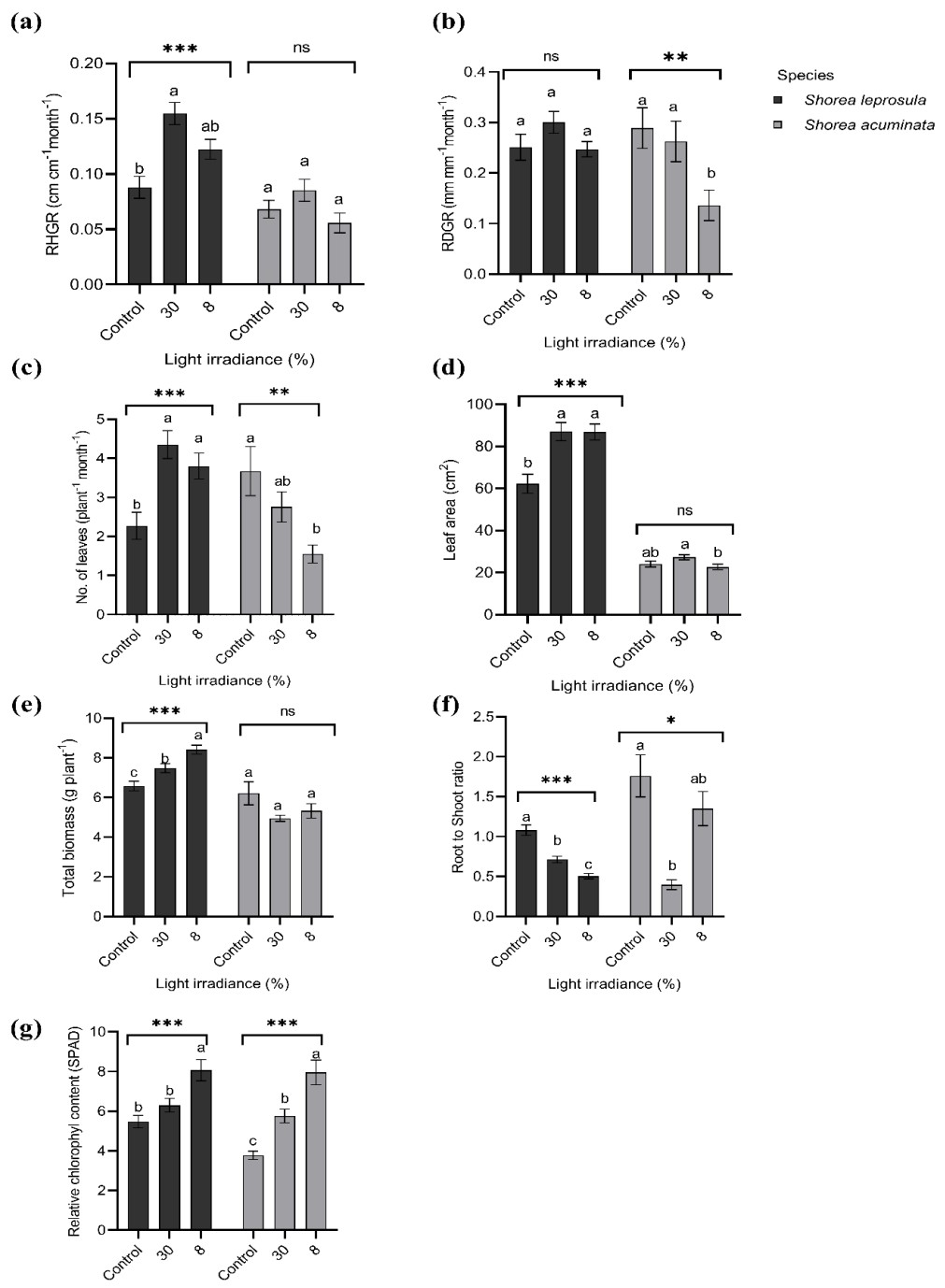

**Figure 4.** The growth performance of *Shorea leprosula* Miq. and *Shorea acuminata* Dyer to different light irradiances; (**a**) relative height growth rate (RHGR) (cm cm$^{-1}$ month$^{-1}$); (**b**) relative diameter growth rate (RDGR) (mm mm$^{-1}$ month$^{-1}$); (**c**) mean number of leaves (plant$^{-1}$ month$^{-1}$); (**d**) leaf area (cm$^2$); (**e**) biomass (g plant$^{-1}$); (**f**) root-to-shoot ratio; and (**g**) relative chlorophyll concentration (SPAD) (mean ± SE). Alphabets denote significant differences between light irradiance treatments with *, **, and *** indicating significant at $p < 0.05$, $p < 0.01$, and $p < 0.001$, respectively; ns = non-significant.

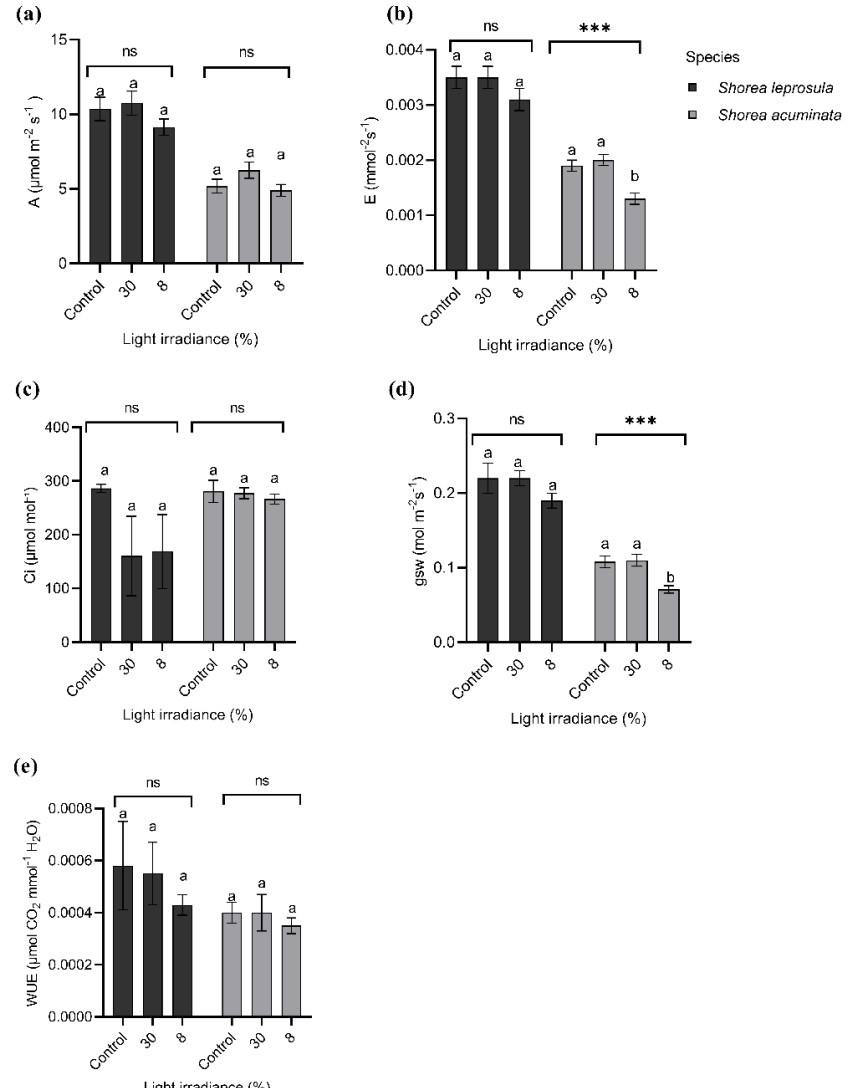

**Figure 5.** *Shorea leprosula* Miq. and *Shorea acuminata* Dyer gas exchange characteristics in different light irradiances (mean $\pm$ SE); (**a**) photosynthesis rate (A, $\mu mol\ m^{-2}\ s^{-1}$); (**b**) transpiration rate (E, $mmol^{-2}\ s^{-1}$); (**c**) intercellular carbon concentration (Ci, $\mu mol\ mol^{-1}$); (**d**) stomatal conductance ($g_{sw}$, $mol\ m^{-2}\ s^{-1}$); (**e**) and water-use efficiency (WUE, $\mu mol\ CO_2\ mmol^{-1}\ H_2O$). Alphabets denote significant differences between light irradiance treatments with *** indicating significant at $p < 0.001$, respectively; ns = non-significant.

### 3.2. Growth and Photosynthetic Responses of S. leprosula and S. acuminata Grown in Different Nutrient Amendments

Similarly, in response to different light irradiances, *S. leprosula* has better growth performance than *S. acuminata* except for the root-to-shoot ratio (Figure 6). The RHGR, RDGR, mean number of leaves, biomass, and RCC for *S. leprosula* showed the highest values with vermicompost amendment compared to the control (no fertiliser). The RHGR, RDGR, and biomass were 36%, 50%, and 6% higher in vermicompost, respectively, when compared to the control. Additionally, the mean number of leaves was twofold higher with the addition of VC compared to the control. The leaf area and RCC showed the opposite trend, with the highest value observed in NPK. The leaf area and RCC were 40% and 20% higher than the control, respectively. The root-to-shoot ratio was approximately twofold lower in VC compared to the control.

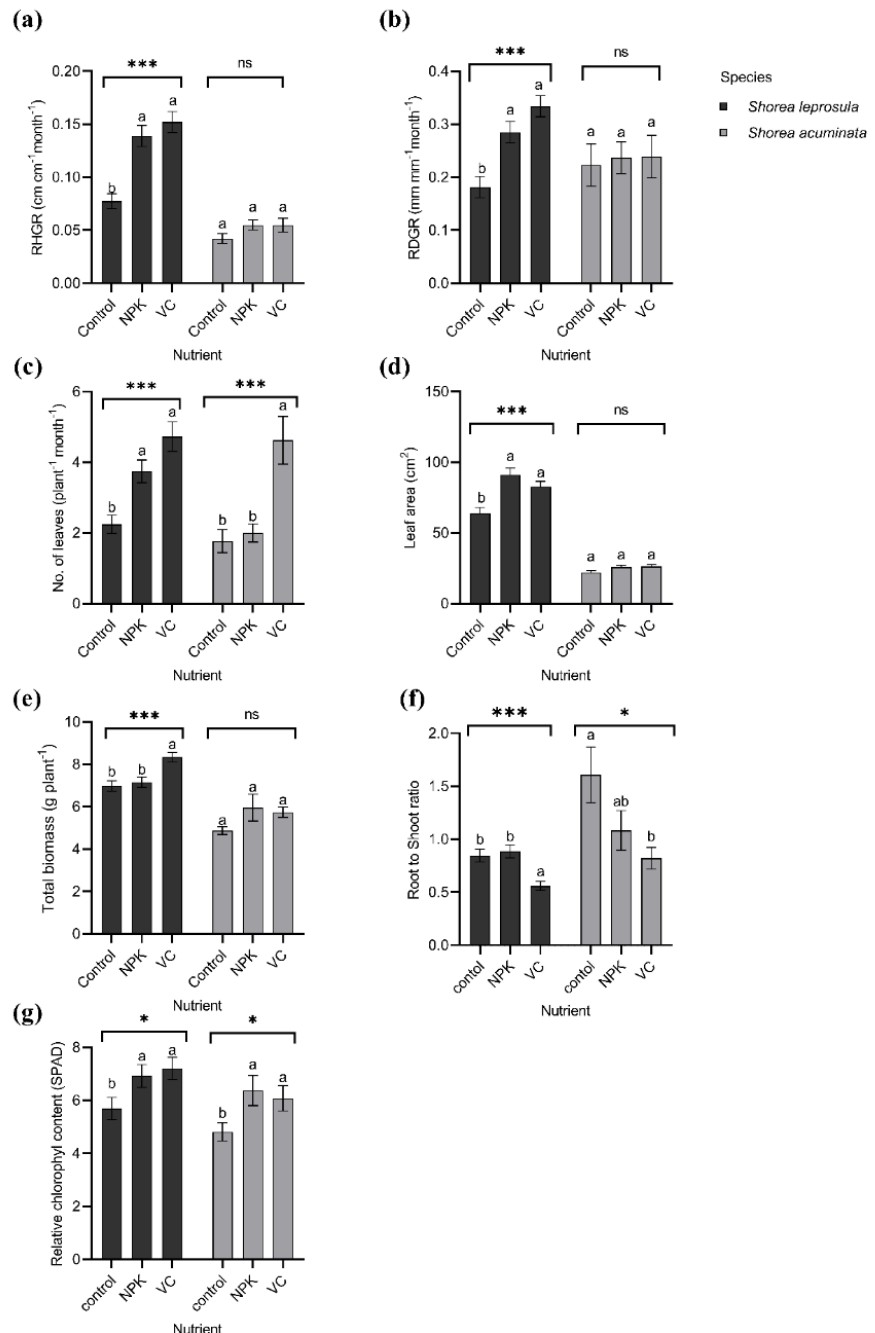

**Figure 6.** The growth performance of *Shorea leprosula* Miq. and *Shorea acuminata* Dyer to different nutrient amendments; (**a**) relative height growth rate (RHGR) (cm cm$^{-1}$ month$^{-1}$); (**b**) relative diameter growth rate (RDGR) (mm mm$^{-1}$ month$^{-1}$); (**c**) mean number of leaves (plant$^{-1}$ month$^{-1}$); (**d**) leaf area (cm$^2$); (**e**) biomass (g plant$^{-1}$); (**f**) root-to-shoot ratio; and (**g**) relative chlorophyll concentration (SPAD) (Mean ± SE). Alphabets denote significant differences between light irradiance treatments with *, and *** indicating significant at $p < 0.05$, and $p < 0.001$, respectively; ns = non-significant.

The growth responses of *S. acuminata* only showed mean number of leaves as the highest value (approximately twofold) in VC compared to the control (Figure 6c). The RHGR, RDGR, and leaf area for both NPK and VC shared the same value and were 27%, 9%, and 17% higher than the control, respectively (Figure 6a,b,d). The RCC and total biomass were higher in NPK with 35% and 20%, respectively, compared to the control (Figure 6e,g). The root-to-shoot ratio was twice as low in VC than the control (Figure 6f).

*Shorea leprosula* has higher leaf gas exchange characteristics compared to *S. acuminata*. The photosynthesis rate (A), transpiration rate (E), and stomatal conductance ($g_{sw}$) were higher in the application of NPK with approximately 55%, 6%, and 15% higher compared to the control, respectively (Figure 7a,b,d). In contrast, WUE was 9% higher with the addition of vermicompost compared to the control (Figure 7e).

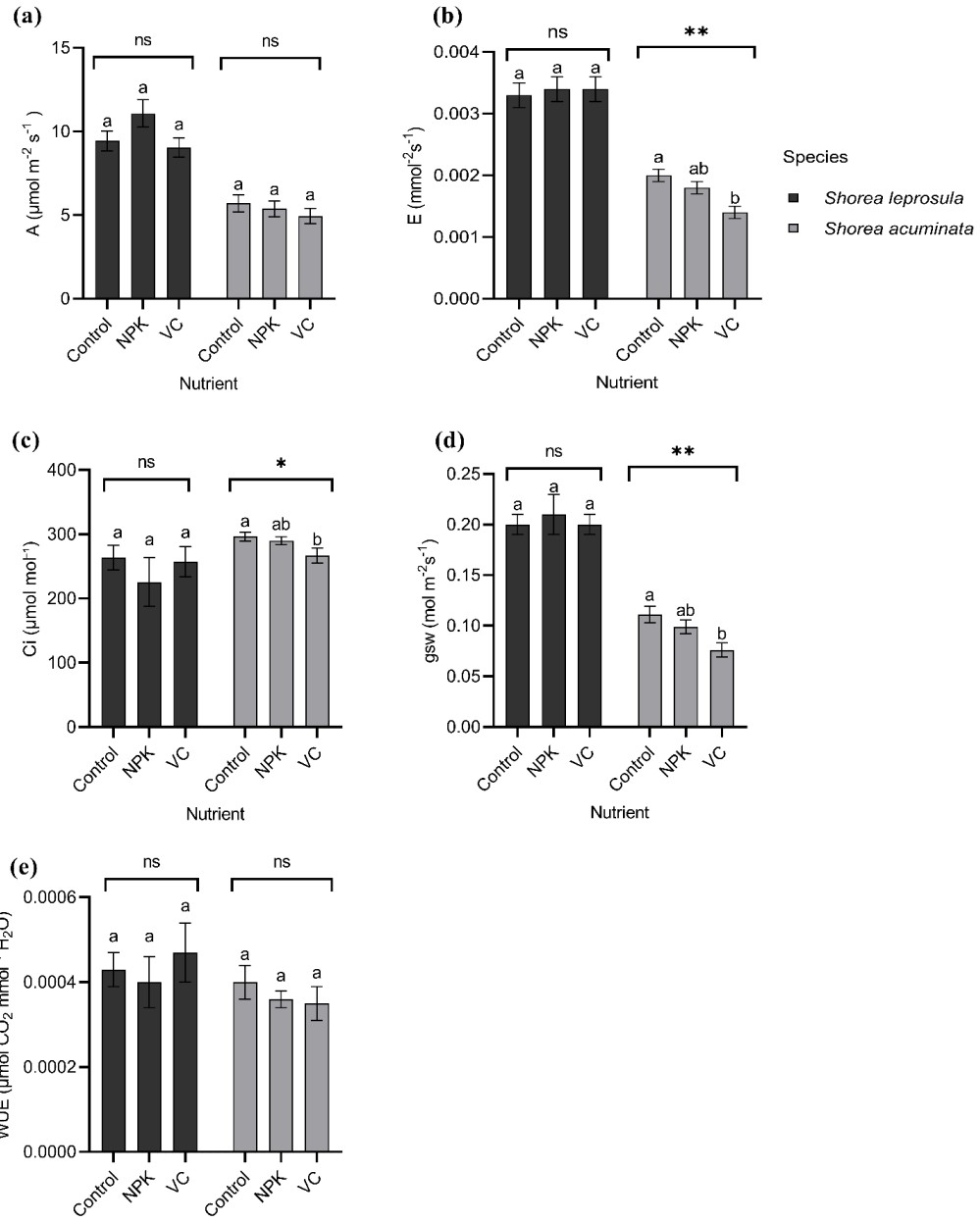

**Figure 7.** *Shorea leprosula* Miq. and *Shorea acuminata* Dyer gas exchange characteristics in different light irradiances (Mean ± SE); (**a**) photosynthesis rate (A, µmol m$^{-2}$ s$^{-1}$); (**b**) transpiration rate (E, mmol$^{-2}$ s$^{-1}$); (**c**) intercellular carbon concentration (Ci, µmol mol$^{-1}$); (**d**) stomatal conductance ($g_{sw}$, mol m$^{-2}$ s$^{-1}$); (**e**) and water-use efficiency (WUE, µmol $CO_2$ mmol$^{-1}$ $H_2O$). Alphabets denote significant differences between light irradiance treatments with *, and ** indicating significant at $p < 0.05$, and $p < 0.01$, respectively; ns = non-significant.

The trend observed in *S. acuminata* was contrary to *S. leprosula*. All the leaf gas exchange characteristics showed a decreased value in VC compared to the control (Figure 7a–e).

The interaction between light and nutrients showed 8% irradiance × VC (8%*VC) obtained the highest total biomass for *S. leprosula* (Figure 8a), while the control treatment C × NF (100% irradiance × no fertiliser) had the lowest total biomass. As for *S. acuminata*, the highest total biomass was obtained using C × NPK, while the lowest was the 30% × NPK (Figure 8b). However, the differences were not significant in *S. acuminata*. The root-to-shoot ratio showed the highest value in C × NPK and C × NF (control), while the lowest root-to-shoot ratio for *S. leprosula* was observed in 8% × VC (Figure 9a). *Shorea acuminata* showed the highest root-to-shoot ratio in the control treatment (C × NF) and the lowest in 8% × NPK (Figure 9b).

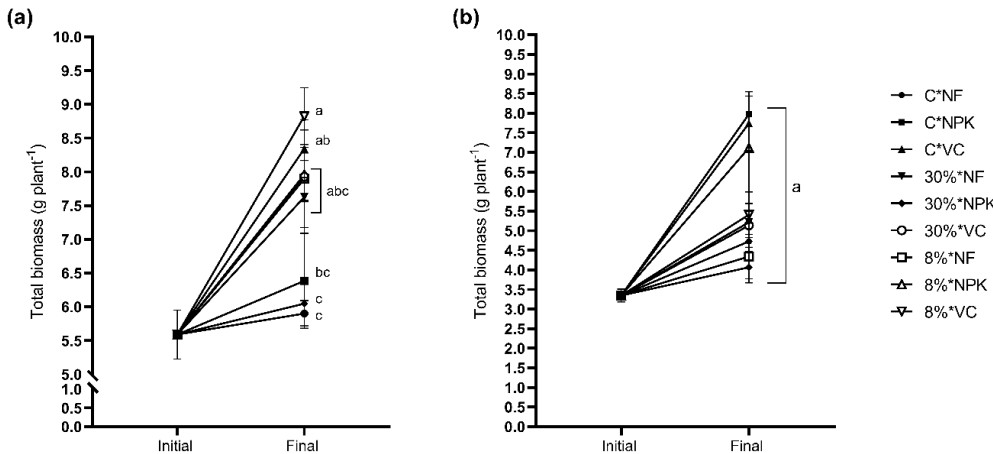

**Figure 8.** Total biomass of *Shorea leprosula* Miq. (**a**) and *Shorea acuminata* Dyer (**b**) in response to the interaction factor, light × nutrient. Alphabets denote significant differences between light × nutrient interaction.

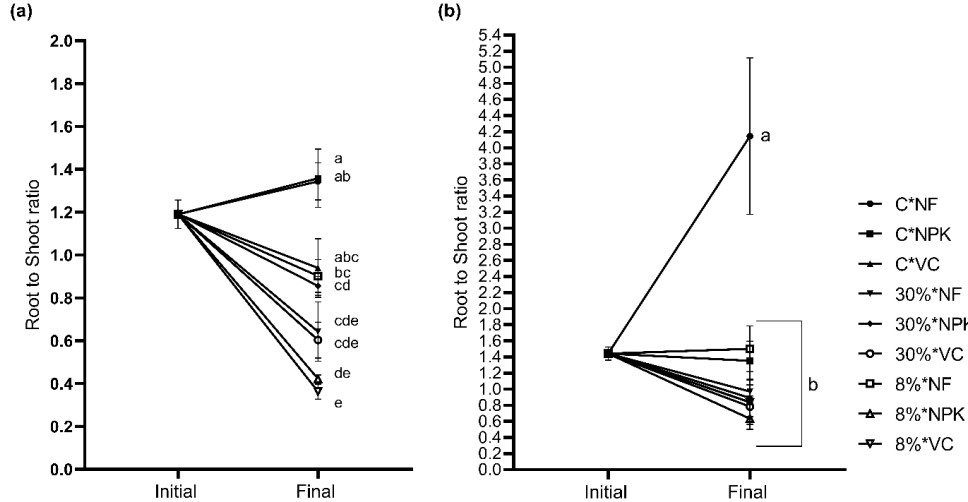

**Figure 9.** Root-to-shoot ratio of *Shorea leprosula* Miq. (**a**) and *Shorea acuminata* Dyer (**b**) in response to the interaction factor, light × nutrient. Alphabets denote significant differences between light × nutrient interactions.

### 3.3. Establishing the Optimum Level of Light Irradiance and Nutrient Amendments

Principal components analysis based on 12 plant traits explained 79.6% of the variance in the first three principal components (Table 3 and Figure 10). The first component (PC1) represented 53.0% of the variability and accounted primarily for RHGR, RDGR, leaf area, biomass, and WUE. The second component (PC2) represented 15.1% of the variance and primarily comprised the mean number of leaves, root-to-shoot ratio, E, A, Ci, and $g_{sw}$ (Table 3). Biplots from PCA analysis showed *S. leprosula* with 30% light irradiance were scattered on the right-hand side of the biplot (Figure 10). The results of PCA indicated that RHGR and leaf area were two key factors (with major positive effects) in PC1 (Table 3).

SLVC30 had higher PC1 scores, indicating better performance for the traits measured than other treatments (Figure 10).

**Table 3.** Variable loading scores of 12 parameters for two *Shorea* species (*S. leprosula* Miq. and *S. acuminata* Dyer) in different levels of light irradiance and nutrient amendment and the proportion of variation of each principal component.

| | Abbreviation | PC1 | PC2 |
|---|---|---|---|
| Relative height growth rate (cm cm$^{-1}$ month$^{-1}$) | RHGR | **0.362** | 0.127 |
| Relative diameter growth rate (mm mm$^{-1}$ month$^{-1}$) | RDGR | **0.294** | 0.016 |
| Mean number of leaves (plant$^{-1}$ month$^{-1}$) | LC | 0.269 | **0.351** |
| Leaf area (cm$^2$) | AREA | **0.377** | −0.047 |
| Biomass (g plant$^{-1}$) | BIOMASS | **0.329** | 0.149 |
| Root-to-shoot ratio | RATIO | −0.246 | **−0.398** |
| Relative chlorophyll concentration (SPAD) | RCC | 0.104 | 0.386 |
| E (mmol$^{-2}$ s$^{-1}$) | E | 0.324 | **−0.406** |
| A (μmol m$^{-2}$ s$^{-1}$) | A | 0.323 | **−0.327** |
| Ci (μmol mol$^{-1}$) | Ci | −0.223 | **−0.288** |
| g$_{sw}$ (mol m$^{-2}$ s$^{-1}$) | GSW | 0.321 | **−0.413** |
| WUE (μmol CO$_2$ mmol$^{-1}$ H$_2$O) | WUE | **0.164** | −0.053 |
| Variability (%) | | 53.0 | 15.1 |
| Cumulative variability (%) | | 53.0 | 68.1 |

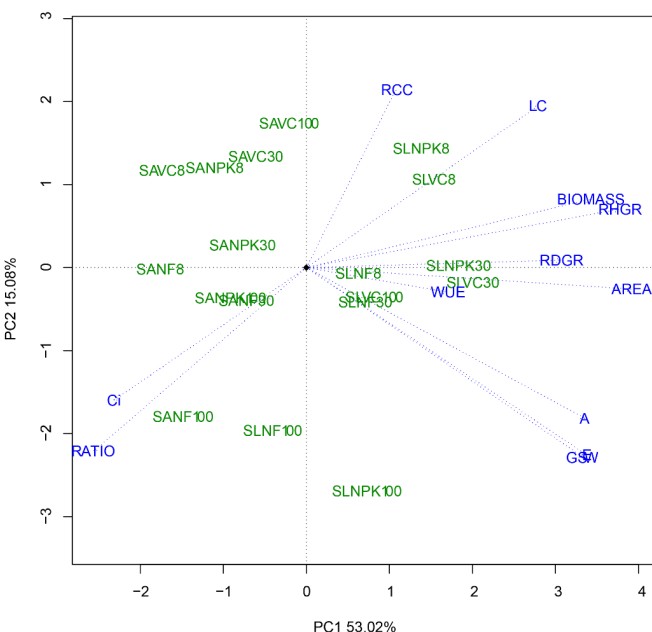

**Figure 10.** Principal component analysis (PCA) biplot for two *Shorea* species, *S. leprosula* (SL) and *S. acuminata* (SA), growing under different levels of light irradiance (8%, 30%, 100%) and nutrient amendments (NF, NPK, VC). Growth responses: relative height growth rate (RHGR) (cm cm$^{-1}$ month$^{-1}$), relative diameter growth rate (RDGR) (mm mm$^{-1}$ month$^{-1}$), mean number of leaves (LC) (plant$^{-1}$ month$^{-1}$), leaf area (AREA) (cm$^2$), biomass (g plant$^{-1}$), Root-to-shoot ratio (RATIO), and relative chlorophyll concentration (RCC) (SPAD). Gas exchange measurements: photosynthesis rate (A, μmol m$^{-2}$ s$^{-1}$), transpiration rate (E, mmol$^{-2}$ s$^{-1}$), stomatal conductance (g$_{sw}$, mol m$^{-2}$ s$^{-1}$), water-use efficiency (WUE, μmol CO$_2$ mmol$^{-1}$ H$_2$O), and intercellular carbon concentration (Ci μmol mol$^{-1}$).

## 4. Discussion

### 4.1. Growth and Photosynthetic Responses of S. leprosula and S. acuminata Grown in Different Light Irradiances

The present study showed that plants grown in 30% irradiance had the highest RHGR and photosynthetic rate (A) compared to plants grown in 8% irradiance and exposed to direct sunlight (control). The growth results were in line with the rate of photosynthesis (A), with the highest value in 30% irradiance. Plants grown in 8% irradiance recorded the lowest

rate of photosynthesis as they received the lowest light irradiance. Lack of light irradiance will cause the limitation of energy sources to undergo photosynthesis [28] as plant growth and the leaf eco-physiological traits are closely related to the light environment [17,29–31], although various environmental factors exist in the forest. Contrary to the current results, a high-growth rate of *Dyera costulata* and *Dipterocarpus baudii* were observed under high-light conditions that correspond to the high photosynthetic rate [32]. This is because each plant has its own optimal light irradiance range to support growth [33]. Light can become a stress factor if supplied too much or too little. The increase in illumination may also cause photoinhibition and decreased photosynthetic efficiency if the input of photons exceeds the plant's photosynthetic capacity [34–37]. Excessive light irradiance will cause over energisation of the photosynthetic apparatus, leading to photoinhibition or even photo-destruction [28]. The plants grown under control showed photoinhibition symptoms with lower growth performance and yellowish leaf appearance, but they had developed a specific mechanism to adapt to changes to survive. Roeber et al. [38] stated that plants exhibit two different systems to perceive environmental light information viz. photoreceptors and chloroplast to cope with oxidative damage under high-light stress as an adaptive mechanism. In addition, we believe the rate of photosynthesis fluctuates within the growth time frame. The initial, mid, and final photosynthetic measurements might have different results as the microclimatic changes also affect the whole process. Usuda [39] also supported that the rate of photosynthesis changes very much during development.

Direct sunlight caused S. *leprosula* and *S. acuminata* to gain the highest root-to-shoot ratio and intercellular carbon concentration (Ci). A high level of intercellular $CO_2$ induces closure of the stomata in the same way as ABA-induced closure [40] which also resulted in lower A under 100% light irradiance in this study. Carbon dioxide diffuses through stomatal pores on the leaf surfaces, altering the intercellular carbon concentration (Ci) [41]. The intercellular $CO_2$ was essential to indicate the $CO_2$ substrate available for photosynthetic assimilation. Additionally, Ci also controls the opening and closing of the stomata. A high concentration of $CO_2$ in intercellular spaces causes partial closure of the stomata during daytime [42]. Once the stomata are closed, external $CO_2$ concentration does not affect stomatal movement.

*4.2. Growth and Photosynthetic Responses of S. leprosula and S. acuminata Grown in Different Nutrient Amendments*

The nutrient application showed positive plant growth performance for both species. Vermicompost (VC) recorded better growth performance than the well-known chemical fertiliser, NPK, but the results were more prominent in *S. leprosula*. The present study showed that the application of vermicompost had a significantly lower root-to-shoot ratio than the control. Despite better morphological features observed with the application of VC, the rate of photosynthesis (A) showed better responses towards NPK.

According to Lynch et al. [43], the reduction in the root-to-shoot ratio is associated with the increment of soil fertility, which displays that the shoot growth was higher than root growth. Blouin et al. [44] mentioned that the addition of vermicompost significantly increased shoot biomass by 78% and root biomass by 57%. The present study showed that organic fertilisers can provide better growth than chemical fertilisers. This result also supports the previous study which proved the use of biostimulants (with low NPK values) can produce the same plant growth promotion effect comparable to chemical fertiliser (high NPK values) application [45]. Vermicompost has been widely used in agriculture to increase crop production [46–49]. Using vermicompost to rehabilitate degraded soil would be the most promising way to enhance soil physicochemical recovery. Lal [50] and Dignac et al. [51] mentioned that the large-scale use of composts is a good way to increase the soil content in organic matter, which is critical for their long-term fertility. Altogether, the growth responses tend to be greater under conditions where plants have access to adequate nutrients [52].

Surprisingly, *S. leprosula* grown in vermicompost application had the lowest A. According to Longstreth and Nobel [53], plant mineral status markedly affects the rate of photosynthesis. Two types of mineral nutrients often limit plant growth. These nutrients are nitrogen (N) and phosphorus (P) which are required in large amounts to their availability in soil [54]. Plants require a certain amount of nutrients to support their growth. The amount could be minimum, optimum, or beyond optimum, which can cause toxicity and later reduce growth and photosynthetic rates. Worse come to worst, plants will eventually die [55,56]. The nutrient content in the control soil might have sufficient amounts of nutrients to support the photosynthesis processes. Thereby, the plants do not show any response toward the additional nutrient. The results were opposed to the results obtained by Wright et al. [57] in tree seedlings, saplings, and poles of *Alseis blackiana*, a lowland tropical forest tree species that showed increased growth by enhancing photosynthesis in response to fertilisation.

### 4.3. Interaction between Species or Factors

*Shorea leprosula* and *S. acuminata* showed different light preferences, with *S. leprosula* exhibiting better growth. Both species were known as high light-demanding and for rapid early growth [58–60], but the present results were less pronounced in *S. acuminata*. Competition between both species for resources might be why both were growing together. *Shorea leprosula* were investing more on height increment, while *S. acuminata* were observed to have higher leaf proliferation. Schmitt and Wuff [61] stated that the presence of taller neighbours will decrease the red or far-red light ratio above the crowns of shorter trees, triggering shoot elongation of shorter trees. This would be a morphological adaptation for *S. acuminata* to anticipate and avoid increasing competitive intensity for light. According to Appanah and Weiland [4], *S. leprosula* has been known as a dipterocarp species tolerant to water stress and a light demander in the early stage of growth. In addition, this species can adapt to a wide range of site distribution [62]. *Shorea leprosula* was also a fast-growing, light hardwood species and high light demander [63,64]. Suzuki et al. [64] also stated that this species requires relatively higher light levels for survival or growth. Lack of research addressing the effect of light on *S. acuminata* as this species was reported to have adaptations similar to *S. leprosula.*

The difference between these two species was easily spotted as *S. leprosula* has a bigger leaf, while *S. acuminata* has a smaller pinnately compound leaf. This feature explains the significant difference between both species leaf areas. Size and shape of leaves are largely genetically controlled, but the developmental flexibility exists even within an individual plant depending on environmental circumstances prevailing during leaf formation [65,66]. Leaves expand to intercept light for photosynthesis, take up carbon dioxide, and transpire water for cooling and circulation [65,67]. The sophisticated sensing mechanisms in plants help determine the available nutrients in soil, atmosphere, and light, thus conveying a response by regulating biochemical processes.

The addition of any resources should increase plant growth because plants adjust the allocation of resources until growth is equally limited by all resources [68]. Both species showed different nutrient acquisition strategies. The nutrient-uptake efficiency influenced both species' growth performances and photosynthetic responses towards nutrients. *Shorea leprosula* showed that the root efficiency might cause better responses in absorbing the available nutrients, and *S. acuminata* does not show any responses with any nutrient amendments. The justification for this scenario might be the types of nutrient storage applied by plants. As mentioned by Chapin et al. [69], there were three types of nutrient storage: accumulation, reserve formation, and recycling. Accumulation of nutrient compounds does not directly promote growth, while reserve formation involves the synthesis of storage compounds from resources that might otherwise promote growth. Recycling storage involves the compounds breaking down and possibly mobilising for later growth.

Competition among plants for resources has long been measured to generate stress for plants and is important for determining the distribution of species [70]. In this study, the competition for resources between species had caused *S. acuminata* to be outcompeted by *S. leprosula* as the growth performance was more pronounced in *S. leprosula*. As Craine and Dybzinski [70] mentioned, the presence of multiple plants in a given volume of soil can induce nutrient stress in a given plant as neighbors acquire limitations. Each plant individual differentially captures a potentially common limiting resource supply. Additionally, the responses of individuals in a mixture also reflects their interactions with their biotic and abiotic environments [71,72].

## 5. Conclusions

The alarming state of the current climate has forced the urge to initiate the rehabilitation of degraded tropical forests. Rehabilitation or restoration should emphasise bringing back the forest service instead of forest cover. The species selection should be prioritised to mitigate climate change by absorbing more carbon and optimising the conditions under which increased photosynthesis can lead to maximal growth increases. In addition, the tree selection should not restrict their tolerance to available light but also the ability to adapt to their microclimate. Light irradiance and nutrient amendment were proven to affect the growth performances of *S. leprosula* and *S. acuminata*, and both species showed different preferences. Plants grown in 30% irradiance and vermicompost applications were considered an ideal condition for better growth performance. Generally, both treatments contributed to an approximately 50% increase in height, diameter, and leaf proliferations compared with the control plants. Vermicompost was widely used and known to bring success in agriculture by providing a better yield, but the application of vermicompost towards dipterocarps tree species was limited. Based on the results, vermicompost application was highly recommended in large-scale dipterocarp plantations to rehabilitate degraded tropical forests as it provides both short-term and long-term benefits. This will conserve dipterocarp species from extinction and improve the productivity of the tropical rainforest.

**Author Contributions:** A.R.S.N.; writing—original draft preparation, W.A.W.J.; writing—review and editing, K.S. and M.N.S.; supervision. All authors have read and agreed to the published version of the manuscript.

**Funding:** This study was funded by Tenaga Nasional Berhad Research (TNBR) research grant (ST-2017-010) and Research University Grants (GUP) (GUP-2018-022) and in collaboration with Terengganu Forestry Department (JPNT).

**Data Availability Statement:** Not applicable.

**Acknowledgments:** The authors would like to thank the anonymous reviewers for their valuable suggestions and comments on the manuscript. Thanks to Jean Wan Hong Yong from Swedish University of Agriculture Sciences for all the valuable insights, especially on site, and Mohd Ikmal Asmuni for the helpful guidance in data analysis.

**Conflicts of Interest:** The authors declare no conflict of interest. The funders had no role in the design of the study; in the collection, analyses, or interpretation of data; in the writing of the manuscript; or in the decision to publish the results.

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
