# Peer review of "Morpho-Physiological Strategies of Shorea leprosula Miq. and Shorea acuminata Dyer in Response to Light Intensity and Nutrient Amendments"

_forests, doi:10.3390/f13111768_

Round 1

Reviewer 1 Report

Minor comments

Title

The lines 2-3, use “nutrient amendments” instead of “nutrients amendments”

Abstract

The lines 19-22, you mentioned abbreviations such as (RHGR, A value, RDGR). The first time you use an abbreviation, you should spell it out completely.

At the end of the abstract, the most important findings and the study's potential contributions should be provided.

1.     Introduction

Line 38, use “have” instead of “has”
Line 41, Change “the increasing demands of resources” into “the increasing demands for resources”

2.     Methodology

The line 100-103, how many seedlings were used in each experimental unit, please?

It would be useful to add photos showing the shape and size of the seedlings that were used in this study.

3.     Results

Line 147-148; change “Table 1. The effects of light irradiance ad nutrients amendment on growth ad photosynthetic characteristics in two Shorea species.” into “Table 1. The effects of light irradiance and nutrient amendments on growth and photosynthetic characteristics in two species of Shorea.”

The format of Table 1 is inappropriate. I recommend dividing it into two tables. The first table should be for the vegetative characteristics and the second for the photosynthetic characteristics.

4.     Discussion

The subheadings in both the Discussion and Methodology sections should be numbered.

Reviewer 2 Report

Manuscript entitled " Morpho-physiological strategies of Shorea leprosula and Shorea acuminata in response to light intensity and nutrients amendments by RAZAK and team.

Overall, the manuscript has very informative and could attract many researchers in the relevant field of forestry and agroforestry. They highlighted that the successfully restoring degraded forest areas depends on seedlings adapting their growth to suit harsh environments.

Pl see below comments-:

Abstract: Well written, add overall recommendation.

Keywords: Ok

Introduction: very short, objective is missing, need updated literature.

M & M section is well written:

Ø  Add map of the study area

Ø  Add high-resolution photographs of study

Ø  Data Analysis – Ok

Result: Overall results is well define, but still in result pl add % changes in different treatments so that it can be comparison.

Figure 2: Missing – pl check

Figure 3 & 4 – Very informative

Figure 2 on page 10 pl see and resolve it at appreciate place

Author mentioned 8 figure in citation and originality is misleading please see.

Figure 8 – need high resolution photographs

Discussion: need major modification with adding latest information, add scientific statements for drawing any conclusion, authors included general statements from other study they should include and correlated with their results so that a proper justification should be draw.

Conclusion: is very short, add quantify data.

Overall, the manuscript has very information,

Reviewer 3 Report

The paper entitled: "Morpho-physiological strategies of Shorea leprosula and Shorea acuminata in response to light intensity and nutrients amendments", discusses the effect of irradiance (8%, 30%, and 100%), nutrient addition (No fertiliser (NF), NPK, and vermicompost  on the the growth and the physiology of two Shorea species, Shorea leprosula and Shorea acuminata.

The paper is of a high interest as it presents new findings related to the morpho-physiological response of these species. However, the presentation of the paper is poor, I recommend considering the paper after major revisions.

* For the data collection section, authors must add some recent and relevant references of the methodologies adopted, namely for the photosynthesis rate, transpiration rate, stomatal conductance, water use efficiency, and the intercellular carbon concentration.

* Figures are in an incorrect order, see 9 - 10 of 16, some figures are missing see figure 5.

** I invite the authors to check the references section (it should respect Forests guidelines) as it should contain more information about each paper, and I take the occasion to invite the authors to cite more recent reports in their intoduction section.

Round 2

Reviewer 2 Report

Now revised version recommended for future process.

Reviewer 3 Report

After consulting the authors responses for all the reviewers assigned, I recommend the publication of the paper at its current form. The authors did a great job in the first round. Well-done.